# 3D Printed Reconfigurable Modular Microfluidic System for Generating Gel Microspheres

**DOI:** 10.3390/mi11020224

**Published:** 2020-02-21

**Authors:** Xiaojun Chen, Deyun Mo, Manfeng Gong

**Affiliations:** School of Mechanical and Electronic Engineering, Lingnan normal university, Zhanjiang 524048, China; deyun_mo@163.com (D.M.); gongmf@Iingnan.edu.cn (M.G.)

**Keywords:** modular microfluidic system, 3D printing, gel microspheres

## Abstract

Integrated microfluidic systems afford extensive benefits for chemical and biological fields, yet traditional, monolithic methods of microfabrication restrict the design and assembly of truly complex systems. Here, a simple, reconfigurable and high fluid pressure modular microfluidic system is presented. The screw interconnects reversibly assemble each individual microfluidic module together. Screw connector provided leak-free fluidic communication, which could withstand fluid resistances up to 500 kPa between two interconnected microfluidic modules. A sample library of standardized components and connectors manufactured using 3D printing was developed. The capability for modular microfluidic system was demonstrated by generating sodium alginate gel microspheres. This 3D printed modular microfluidic system makes it possible to meet the needs of the end-user, and can be applied to bioassays, material synthesis, and other applications.

## 1. Introduction

The reconfigurable microfluidic system are connected by basic module components together to form an integrated microfluidic system, which could achieves specific functions of biochemical analysis, such as emulsion generation [1], multi-organ-chips [2], gradient generation [3], and biochemical analysis [4]. Existing systems often use a monolithic approach, where chemical reactors, sensors, valves, pumps, and detectors are integrated on a single chip. The typical fabrication methods of monolithic microfluidic systems, which include soft lithography, hot embossing, and femtosecond laser writing, are time consuming and often expensive. Additionally, the complicated fabrication processes would require more attention to the quality control [5]. Any part of failure of a monolithic microfluidic system may require rebuilding the entire system, which will result in long development time and incur substantial costs. The modular design approaches are an effective method to address this integration problem. The advantage of a modular system is made of the assembly of basic modular components, and each module can be designed and tested separately before connecting them together to form a larger system.

Modular architecture using prefabricated microfluidic components could be easily assembled, disassembled, reconfigured, and assembled again. Nevertheless, reconfigurable modular microfluidic systems have brought challenges to ensure leak-free fluidic interconnections between connected microfluidic modules after their assembly. Hsieh et al. presented an advanced Lego^®^-like swappable fluidic module concept to achieve fully portable, disposable fluidic systems [6]. This module design had self-aligning structures on both the male- and female-type Lego^®^-like for attaining the improved sealing at block junctions. Bhargava et al. demonstrated an approach to microfluidic device design based on discrete elements. These components were connected by a convex block embedded in a concave block to complete the assembly of the entire microfluidic system [7,8,9]. The aforementioned microfluidic devices were adopted self-aligning structures for attaining the inter-block sealing at block junctions. The instability of the module-to-module fluidic interconnects required to be considered, easily led to fluid leakage, to operate the integrated device under high pressure. In addition, Rheea and Burns proposed a standard set of modular microfluidic assembly block, using pre-fabricated polydimethylsiloxane (PDMS) blocks [10]. This approach required an additional glue-like, UV curable adhesive, as well as PDMS mixture to connect components. Lee et al. proposed an advanced fabrication and assembly method for modular microfluidic devices using rubber O-rings and metal pins interconnects [11]. From the user’s point of view, modular microfluidic devices allowed non-expert users to assemble fully customizable microfluidic devices in minutes. However, these module-to-module or world-to-chip fluidic interconnects are required to be strengthened by additional bonding and sealing processes, which will result in a long development time and incur substantial costs.

Recently, 3D printing technology have been applied to fabricate miniaturized and complicated devices, which are of high structural complexity and design flexibility [12,13,14,15,16]. In this work, we propose an advanced reconfigurable modular microfluidic system employing 3D printed modules with assorted channel geometries that can be easily assembled to create complex, modular, and reversible in three dimensions. The microfluidic system consists of two basic functional components: A screw fastener and an assembly module. Each assembly module has an own unique function (such as inlets, outlets, channels, valves, pumps, mixers, and reservoirs) that is connected together and bonded to form a multi-function microfluidic system using screw interconnects. Furthermore, screw connectors are inserted into each microfluidic module’s threaded port to eliminate fluid leakage and enable high pressure actuation. It offers a promising way to realize a larger integrated microfluidic system capable of sophisticated functionalities.

## 2. Materials and Methods

### 2.1. Materials and Instruments

Ethanol, dye solutions (red and blue) and food grade mineral oil were purchased from Sinopharm Chemical Reagent Co., Ltd. The dye solution was dissolved with deionized (DI) water. Sodium alginate powder and calcium chloride powder were obtained from Sigma-Aldrich Corporation (Guang Dong, China). Materials for the out phase were 2% (*w*/*w*) calcium chloride. The pre-gel aqueous phases were 2% (*w*/*w*) sodium alginate. The PDMS was obtained from Dow Corning (Midland, MI, USA). The PDMS was heat cured at a temperature of 100 °C for 20 min in a vacuum drying oven. Fine sandpaper (P1500 and P2000) was purchased from Xiamen Green Reagent Glass Instrument Co., Ltd. (Xiamen, China). An optical microscope (Mitutoyo MF-U, Chongqing Aote Optical Instrument Co., Ltd., Chongqing, China) was used to take images of the mixer. A laser confocal scanning microscope (Produced by Carl Zeiss AG, OLS1200, Carl Zeiss AG, Oberkochen, Germany) was used to measure the surface roughness of model parts. The COLOR intensity was measured using ImageJ. The experimental process was observed by a CCD camera (UI-2250SE-C-HQ, Shanghai Lingliang Optoelectronics Technology Co., Ltd., Shanghai, China).

### 2.2. Fabrication of Basic Functional Components

To build the 3D models for all of the 3D microfluidic components in this work, we used the computer-aided design (SolidWorks) software. We assembled the 3D models of the ports, components, and systems within the software, and exported the assemblies to the STL format-a standard file type for 3D printers. All modules were directly fabricated by a 3D printer (ProJet^®^D3510 SD, 3D Systems). This printer is based on the multi jet modeling (MJM) technology using a print head that jets the photopolymer (VisiJet^®^ Crystal, 3D Systems) and the waxy support material (VisiJet^®^ S300, 3D Systems) layer by layer.

The assembly modules were carefully selected from well-known standard component as shown in Figure 1a,b. The assembly modules, which were designed to a standard, cubic geometric footprint (with a size of 10 by 10 by 10 mm^3^), comprised functional elements (assembly module) as well as inlet/outlet modules (screw fastener) for world-to-chip fluidic interconnects. The internal channels of these module were designed into a square cross-section with 0.6 mm side length. Each module had different functionality, which acts as basic building units to construct functional microfluidic device. A multi-function microfluidic system was assembled form several modular components (Figure 1c). The screw fastener acted as module-to-module or world-to-chip fluidic interconnects. The spacers were cylindrical with screw pins at opposite sides. The modules were assembled by interlocking screw pins connectors for each module. Following the 3D printing process, the modules were placed in the convection oven at 80 °C to remove the wax layers. When the wax was fully melted, its residue was first removed in a hot oil bath and was then rinsed by ultrasonication in a water bath that contained detergent. The modules were then further rinsed by ultrasonication in a deionized water bath to ensure that the detergent remains were fully removed, and that each procedure was conducted for at least 30 min.

## 3. Results and Discussion

### 3.1. Surface Treatment and Leak Testing

The printed microfluidic channels and their surfaces were observed with an optical microscope, and typical images are shown in Figure 2a. The chip surface quality shown the traces of resin material layers by layers. The surface roughness of the 3D printed module assembly was 2.161 μm. In order to obtain optical transparency, the 3D printed modules were repeatedly polished by two types of fine sandpaper to form a smooth surface. Additionally, the film layer of PDMS on the surface was coated to make the chip more transparent. The surface roughness of the modular component after surface modification was 0.595 μm. Figure 2b demonstrated a modular microfluidic system after polishing and PDMS coating process. The excellent surface quality of the chip was achieved for easier observation of the experiment results. Therefore, surface polishing and PDMS coating were a very effective method to obtain high surface quality. Figure 2c showed that the internal channel structure still appears blurry. An optical microscope described the rough internal flow channel structure. To obtain a highly transparent microchannel, it was also possible to apply PDMS solution inside. Evenly coating the PDMS solution inside the microchannel could enhance the transparency. The key was to prevent the PDMS solution from blocking the internal microchannel.

Previous literature [17] reported that a wet etching technique was used to treat internal surface roughness of microchannels. Ethanol-diluted acetone solution was circulated in an ultrasonic bath to etch the chip surface. This method significantly reduced the surface roughness of the microchannel without deformation, and the surface roughness after processing was less than 10 nm. Chemical treatment of the internal channels was also an effective method to obtain transparency. Photosensitive resin materials were highly sensitive to organic reagents. Placing the printed module assembly in acetone will dissolve, and high concentrations of methanol and isopropanol (IPA) will whiten the material of the module assembly [18].

To check the capability of screw fastener, we performed three different scenarios such as (a) with using embedded connection [9] (Figure 3a), (b) untightened thread, and (c) tightened thread (Figure 3b). All modules were used digital pressure gauges for leakage tests. The serious level of leakage of bubbles was observed under embedded connection and untightened thread (Figure 3c,d, Appendix A). In contrast, the tightened thread modules could be held up to 500 kPa (Figure 3e, Appendix A). In addition, the Young’s modulus of the post-cured clear resin was 2.7 GPa, which is much larger than that of PDMS. Therefore, our devices, printed with this material, were capable of withstanding higher liquid flow rate and input pressure. Compared with the previous literature (Table 1), the method of screw connection was simple and fast. It could be disassembled and reassembled repeatedly to build different functions of novel microfluidic system. It is widely applied in biochemical fields such as fluid mixing and droplet generation. Importantly, the threaded connection provided high fluid pressure, avoiding fluid leakage problems, and expanding its application.

A reconfigurable modular microfluidic system, comprising a mixer channel and various microfluidic modules as well as inlet/outlet modules for world-to-chip fluidic interconnects, was fabricated and used to demonstrate its reconfigurability to build various integrated microfluidic systems by simply and reversibly assembling various modules together. Three different configurations of a mixer channel modular microfluidic system were built using two (Figure 4a), three (Figure 4b), and four (Figure 4c) outlet channel modules, and two inlet modules. The assembly described in Figure 4a,b were modeled as an equivalent circuit consisting of two branch resistors R ($R=R_{struct}+R_{ref}$) and Rs ($R_{s}=R_{struct}+R_{select}$) grounded by two water reservoirs and terminated by outlet resistor Ro (Figure 4d,e). Each branch was designed to differ by only a reference (*R_ref_*) and selected (*R_select_*) component resistance, while having identical support components resulting in equal structural resistance (*R_struct_*). The volumetric mixing ratio M of streams combined in the outlet resistor was predicted by nodal analysis to have simple dependency on only the selected, reference, and branch structural resistances (Equation (1)) [20].
(1)$$M=\frac{R_{struct}+R_{ref}}{R_{struct}+R_{select}}$$

### 3.2. Reconfigurable Modular System Demonstration

A simple modular microfluidic system was built using one mixer module, a T type module, a flow channel, and three inlet/outlet modules (Figure 5a). To demonstrate the 3D mixing mechanism, numerical simulation using finite element analysis software (COMSOL Multiphysics 5.1) was performed to simulate the mixing effects of microfluidic system with straight-channel modular and a mixer channel modular (Figure 5b). Two streams of solution with a relative species concentration of 1 mol/L and 0 were injected into the mixer module through the inlets; the concentration fields were obtained by solving the incompressible Navier–Stokes and convection diffusion equations in the stationary mode. In Figure 5c, the cross-section (terminal position) of the microfluidic system with the mixer modular shows a uniform distribution of the concentration gradient after passing through 30 mm in length. The normalized concentration shows that the spiral mixer was evenly distributed compared to the straight channel. The microfluidic system with a mixer can achieve 99.7% mixing efficiency at the terminal, while the straight-channel microfluidic system could only achieve 88.3% mixing efficiency. Red and blue color food dye solutions were used to test the modular microfluidic system. Each food dye solution was pumped into the modular microfluidic system at ~20 µL/min by micropump. The dye solution did not have sufficient time to mix through laminar diffusion; hence, the different unmixed streams (a clear fluid interface) could be seen after they pass through the straight-channel (Figure 5e). However, experiments were performed at the same flow rate in a microfluidic system with a mixer (Figure 5d). As the diffusion of dye solution had occurred and the fluids were fully mixed in the designed length. The experimental values well match the simulated ones.

As a biological hydrogel, sodium alginate was usually encapsulated cells or other biomolecules. Due to its good gelling properties, sodium alginate has been widely used in the field of biomedicine [21]. Sodium alginate was a typical example of a cross-linked hydrogel, which can react with divalent cations to form a hydrogel such as calcium, copper, and iron. To demonstrate the versatility of a modular microfluidic system. A versatile droplet generation microfluidic system had assembled to generate single droplet and dual droplets (Figure 6a). Sodium alginate passed through two branch channels as a carrier phase solution; calcium chloride passed through the main channel as a cross-linking phase solution. Two screw pumps modules in the microfluidic system were used to control the input of two branch solutions. When the sodium alginate was cross-linked with the calcium chloride solution, calcium ions diffused from the outside to the inside, and finally formed the sodium alginate hydrogel (Figure 6b). When one of the threaded pumps in the system was turned off, a single droplet mode was formed. Figure 6c illustrated the formation of sodium alginate droplets (Appendix A). The formation rate of sodium alginate droplets was mainly related to the diffusion rate, concentration of calcium ions, and the concentration of sodium alginate. When the flow rate of the carrier phase solution was 0.5 mL/h, the relationship between the length, generation frequency of the alginate gel microspheres and the flow rate of the calcium chloride aqueous solution was obtained. As the flow rate of the calcium chloride solution increased, the length of the sodium alginate microspheres decreased. More sodium alginate microsphere gel could be produced as shown in Figure 6d. Finally, the dual droplet generation mode by controlling two screw pump modules was demonstrated (Figure 6e, Appendix A). The droplet patterns spaced with red and blue can be generated by adjusting the flow rate of the red solution and the blue solution (Figure 6f). Such modular microfluidic systems provided flexibility and versatility to manipulate micro-flows for enhanced and extended applications. Furthermore, the ability to build a reconfigurable modular microfluidic system would be an advantageous platform to substantially enhance design flexibility and improve the system performances of various bio-chemical processes.

## 4. Conclusions

In this study, a sample library of standardized components and connectors manufactured using 3D printing was developed. The variety designs of modules were firmly connected using screw fastener. The screw connection perfectly sealed the interconnection between the modules more firmly and prevented any solution leakage. The module assembly and reconstruction are suitable expand microfluidics to non–expert users. Furthermore, the integrated microfluidic device was also applied as multi-analyte mixing and multi-emulsion generation. This modular microfluidic system has a tremendous potential for realizing mass-production complexes and multiplex systems for the commercialization of the microfluidic platform.

## Figures and Tables

**Figure 1 micromachines-11-00224-f001:**
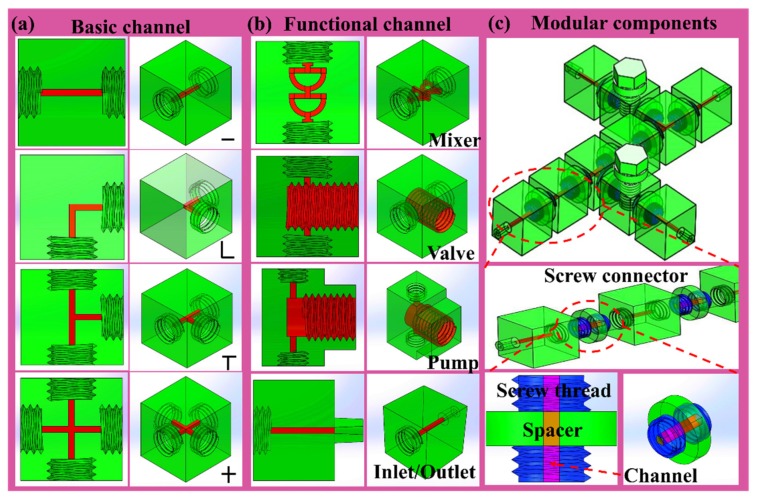
Schematic illustration of 3D printed microfluidic modular components. (**a**) basic microchannel unit. (**b**) functional microchannel unit. (**c**) multifunctional microfluidic system. The device consists of multiple basic channels and functional channels using screw connector.

**Figure 2 micromachines-11-00224-f002:**
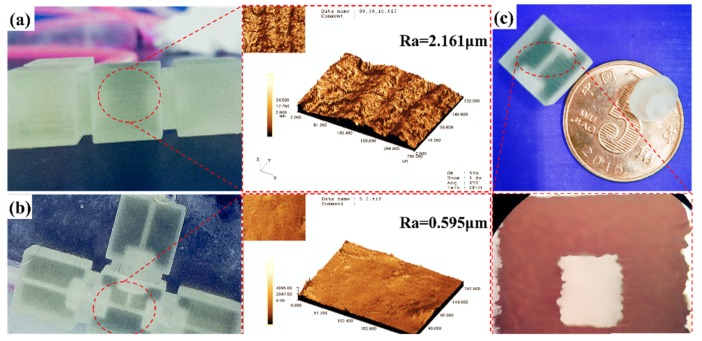
Micrographs of the chip surface quality. (**a**) surface quality and surface roughness before treatment. (**b**) surface quality and surface roughness after polishing and polydimethylsiloxane (PDMS) treatment coating. (**c**) image of 3D printed module components.

**Figure 3 micromachines-11-00224-f003:**
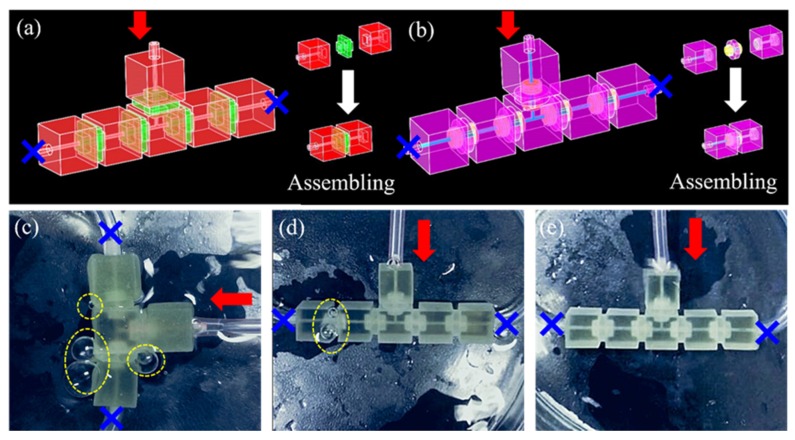
Connectivity of the modular microfluidic system. (**a**) schematic of modular assembly of embedded connections. (**b**) schematic illustration of modular assembly with threaded connections. (**c**) leakage experiment of embedded connected microfluidic system. (**d**) leakage experiment of threaded connected microfluidic system (untightened thread). (**e**) leakage experiment of threaded connected microfluidic system (tightened thread).

**Figure 4 micromachines-11-00224-f004:**
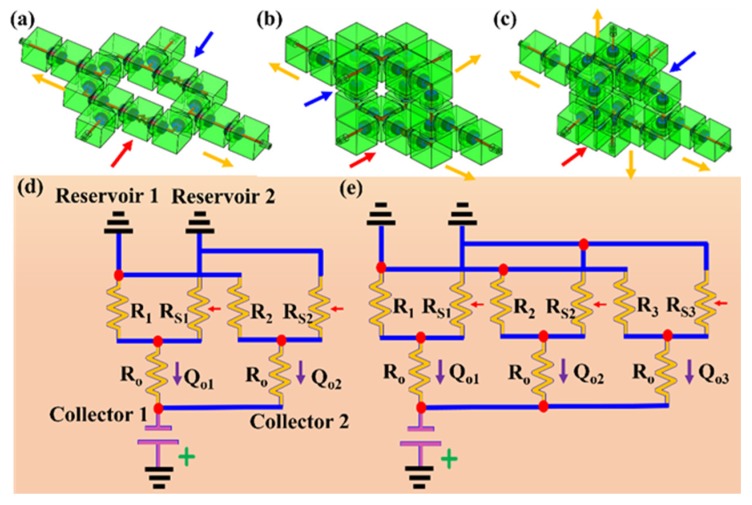
The tunable mixer with (**a**) two, (**b**) three, and (**c**) four outlets. (**d**,**e**) comparison of equivalent circuit models for two- and three-outlet parallelized configurations of the tunable mixer system.

**Figure 5 micromachines-11-00224-f005:**
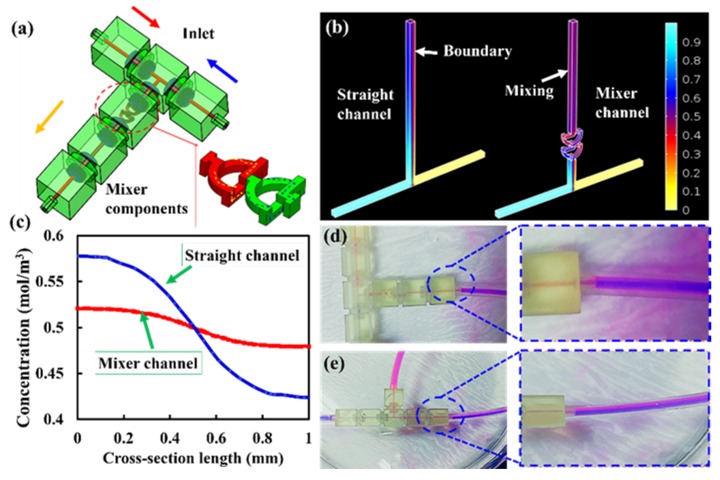
(**a**) schematic illustration of microfluidic system assembly with mixed channel module. (**b**) simulation of mixing effects for straight channel and mixer channel modules. (**c**) normalized concentration of the outlet for the straight channel and mixer channel modules, respectively (the terminal of the microchannel). (**d**,**e**) mixing effects of fluids in straight channel and mixer channel modules.

**Figure 6 micromachines-11-00224-f006:**
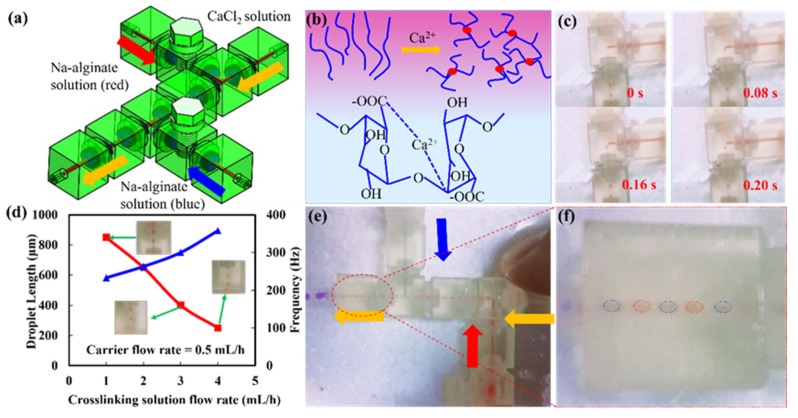
(**a**) modular microfluidic system for droplet generation. (**b**) Ca^2+^ cross-linking to prepare sodium alginate gel microspheres. (**c**) diagram of microsphere generation process. (**d**) the relationship between the length of the gel microspheres and the flow velocity, and the relationship between the flow velocity and the frequency of microsphere generation. (**e**) images of double emulsions. (**f**) Images of double emulsions.

**Table 1 micromachines-11-00224-t001:** Comparison of modular microfluidic systems.

References	Connection Type	Manufacturing Characteristics	Fluid Pressure
Bhargava et al. [7,8,9]	Square interface embedded, comprising spacer and connector	3D printed module, male−male connector aligned with female-type port, reversible strong	As high as 200 mL·h^−1^
M. Rhee and M. A. Burn [10]	UV-curable glue bonding	PDMS coated glass substrate using the curing agent as the adhesive, reversible weak	40 kPa
Lee et al. [11]	Insert connection, comprising rubber O–ring and metal pins	3D printed module, concave and convex cone–shaped features, require auxiliary components. reversible strong	Up to 200 kPa
Yue [19]	Magnetic interconnects, comprising magnets and sealing gaskets	3D printed module, simple, require auxiliary components	6.8 kPa
**Our work**	**Screw connector**	**3D printed module, threaded connection, simple, reversible strong**	**Up to 500 kPa**

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
