# Peer review of "3D Printed Reconfigurable Modular Microfluidic System for Generating Gel Microspheres"

_micromachines, 2020, doi:10.3390/mi11020224_

Round 1

Reviewer 1 Report

In the manuscript, the authors present 3D printed microfluidic blocks that can be assembled individually. The authors characterize the systems and apply for creating gel microspheres. Although this paper demonstrates the technical improvement of the connections and leaking of fluids, I don’t see the novelty of this approach and significant contributions to the communities. I would recommend a significant revision of this manuscript. Here are my corrections and comments.

Line 26-27: In terms of integration and functional efficiency of pumps, valves and sensors, the modular microfluidic systems are more challenging than the monolithic approach, which is well established. The authors should discuss more about this point.

Line 53: Why 3D printing technology is useful/better for this specific application.

Line 84-84: No need this information. Otherwise, it has to be compared with other approaches.

Line 95-97: It needs to be corrected in English.

Line 89-111. These paragraphs do not belong to the result section. It should be a device fabrication section.

Line 115: How these modules are polished?

Line 121: The channel structure is still blurry. The authors have to improve the method and show good devices that is clear. It is very important to have transparent devices for imaging and analysis.

Line 153: The Supporting Information MovieS3 shows the device is still leaking. The authors should demonstrate devices without leakage.

Line 188: The authors should characterize the mixer, e.g, flow rate vs. mixing, comparison discussion with other systems. In Figure 5C, the Y-axis should be normalized value and Figure 5D, the laminar flow interface is not clear. May use a high magnification objective to capture images.

Also, the author should show the relationship between simulation and experimental results.

Line 228: The Supporting movie S5 shows dual drops. However, it seems drops are coalescing downstream of the channel. Also Fig 6 e and f, the image quality should be improved.

References are not well organized: lines 256, 257, 268, 275, 288 and so on.

For the practical applications of this system, I would recommend that the authors demonstrate particles/cell encapsulations with this approach.

Reviewer 2 Report

I find the manuscript interesting and important, but I feel that sizable modifications are needed to reach the level required for publication in Micromachines.

I will go through requests and recommendations line by line.

83: no explanation is given about the chemical and physical properties of VisiJet polymer: what is it chemically ? What chemicals can it tolerate ? Is it hydrophobic ? Is it porous ?

83: term "sacrificial material" is used but on line 108 the term 2wax" is used instead. Please stick to your terminology.

85: printing time: "the whole printing process": does this mean a single module ? Module printing time would make most sense, as systems can be very different.

115: "were polished by fine repeatedely". This sentence is not complete, and needs to be expanded.

116: the application of PDMS for surface smoothing is very unclear. Please give details.

132-139: not clear if this paragraph discusses new results by the authors or some old literature data. Please clarify.

170-173: it would be important to give numerical results. Give component dimensions, flow rates, mixing ratios etc. in a way that an interested reader can check the calculations. It is unclear what information is from ref 19, or are the results discussed new results by the authors.

177, 184: nothing is revealed about the simulation software.

186 "liquid water with relative species concentrations of 1 and 0". Please use standard chemical formulas and concentration nomenclature.

192: please define mixing efficiency

Page 6 as a whole: channel size is not reported.

Figure 6: I feel that a big schematic figure about gel formation would be more useful than many tiny figures of inadequate resolution. Especially c) is too small, and also the insets in e).

What is droplet length ? There must be some mention of channel dimensions.

The most serious problem with the manuscript is lack of discussion. In the introduction existing literature is nicely presented, but in the "Results and discussion" section there is no discussion. The authors should take main ideas and results from references 6-10, and make direct comparisons: how did their new system compare with old results in terms of assembly time, leak tightness, pressure tolerance, sample volumes, dead volumes... Only this way readers can appreciate the advances made in new work. It also helps to raise the most unique points  up front, which makes abstract and conclusions stronger, when absolute novelty can be claimed with stronger evidence.

Round 2

Reviewer 1 Report

The authors have corrected and modified the manuscript based on the comments. However, the experiments have not been conducted based on the comments and still, the image quality is not good enough for publication in Micromachine.

Line 124: Figure 2 (b) and (c) cannot be seen and are not printed well. The channel structure is still blurry. The authors should demonstrate a method that can clearly show the channel structures. There is no reason to take this method that is not feasible to see the microchannels and further applications.

Line 177: The authors show the results of mixing and comparison study. To have a better understanding of the mixing of this system, the authors should perform systematic experiments. Flow rate vs. mixing, distance vs. mixing (Chaotic Mixer for Microchannels, Stroock et al., Science, 2002). It’s very surprising that without the T-mixer, it can be achieved 88% mixing efficiency. What is Re number? How does intensity is measured? The authors should use and do the experiments with fluorescent dye to measure the intensity.

Line 204: Figure 6 e and f, the image quality should be improved. It is still the same. Unless I don’t recommend this approach publishing in the journal of Micromachine.

Reviewer 2 Report

I am suggesting publication of the revised manuscript.

Author Response

We appreciate the reviewer for the time and effort for reviewing original manuscripts. Your comments are all valuable and very helpful for revising. We have uploaded a copy of the original manuscript with all the changes highlighted using the track changes mode in MS Word and a revised manuscript with the correction sections red line marked for easy reference. Please let us know if you have any questions about this revision. Thanks.

Round 3

Reviewer 1 Report

I accept the manuscript.